# Multiple systems in macaques for tracking prediction errors and other types of surprise

**Jan Grohn** [1] *, **Urs Schüffelgen** [1], **Franz-Xaver Neubert**[1], **Alessandro Bongioanni** [1], **Lennart Verhagen** [1,2], **Jerome Sallet** [1,3], **Nils Kolling** [1,4‡], **Matthew F. S. Rushworth** [1‡]

**1** Wellcome Integrative Neuroimaging (WIN), Department of Experimental Psychology, University of Oxford, Oxford, United Kingdom, **2** Donders Institute for Brain, Cognition and Behaviour, Radboud University, Nijmegen, the Netherlands, **3** Univ Lyon, Université Lyon 1, Inserm, Stem Cell and Brain Research Institute U1208, Bron, France, **4** Wellcome Integrative Neuroimaging (WIN), Oxford Centre for Human Brain Activity (OHBA), University of Oxford, Oxford, United Kingdom

☉ These authors contributed equally to this work.
‡ These authors also contributed equally to this work.
* jan.grohn@psy.ox.ac.uk

## Abstract

Animals learn from the past to make predictions. These predictions are adjusted after prediction errors, i.e., after surprising events. Generally, most reward prediction errors models learn the average expected amount of reward. However, here we demonstrate the existence of distinct mechanisms for detecting other types of surprising events. Six macaques learned to respond to visual stimuli to receive varying amounts of juice rewards. Most trials ended with the delivery of either 1 or 3 juice drops so that animals learned to expect 2 juice drops on average even though instances of precisely 2 drops were rare. To encourage learning, we also included sessions during which the ratio between 1 and 3 drops changed. Additionally, in all sessions, the stimulus sometimes appeared in an unexpected location. Thus, 3 types of surprising events could occur: reward amount surprise (i.e., a scalar reward prediction error), rare reward surprise, and visuospatial surprise. Importantly, we can dissociate scalar reward prediction errors—rewards that deviated from the average reward amount expected—and rare reward events—rewards that accorded with the average reward expectation but that rarely occurred. We linked each type of surprise to a distinct pattern of neural activity using functional magnetic resonance imaging. Activity in the vicinity of the dopaminergic midbrain only reflected surprise about the amount of reward. Lateral prefrontal cortex had a more general role in detecting surprising events. Posterior lateral orbitofrontal cortex specifically detected rare reward events regardless of whether they followed average reward amount expectations, but only in learnable reward environments.

**Data Availability Statement:** Data and code associated with this project are available at doi.org/10.5281/zenodo.3993116.

## Introduction

Animals, including humans, learn from the past to predict the future. This enables them to adjust to their environment and is critical for survival. One type of prediction that animals make concerns the reward value of potential choices that might be taken [1, 2]. After the choice and the outcome is experienced, the correctness of the original expectation is evaluated. When

**Funding:** Funding for this work was provided by Medical Research Council (https://mrc.ukri.org/) grants MR/P024955/1 (MFSR, JS, and NK), G0902373 (MFSR), MR/K501256/1 (JG), MR/N013468/1 (JG), Wellcome Trust (https://wellcome.ac.uk/) grants 203139/Z/16/Z (MFSR and US); WT100973AIA (MFSR); WT101092MA (MFSR and JS), 105651/Z/14/Z (JS), Christ Church College, University of Oxford (https://www.chch.ox.ac.uk/) (NK), St John's College, University of Oxford (https://www.sjc.ox.ac.uk/) (JG), the Biotechnology and Biological Sciences Research Council (https://bbsrc.ukri.org/) grant BB/R010803/1 (NK), and the Clarendon Fund, University of Oxford (http://www.ox.ac.uk/clarendon) (FXN). The funders had no role in study design, data collection and analysis, decision to publish, or preparation of the manuscript.

**Competing interests:** The authors have declared that no competing interests exist.

**Abbreviations:** ANTs, Advanced Normalization Tools; EPI, echo planar imagingFEAT, fMRI Expert Analysis Tool; FLAME, FMRIB's Local Analysis of Mixed Effects; fMRI, functional magnetic resonance imaging; FSL, FMRIB Software Library; GLM, general linear model; GLME, generalized linear mixed-effect model; GRE, gradient-refocused-echo; HRF, hemodynamic response function; lPFC, lateral prefrontal cortex; MrCat, Magnetic Resonance Comparative Anatomy Toolbox; PE, prediction error; plOFC, posterior lateral orbitofrontal cortex; ProM, proisocortical motor cortex; ROI, region of interest; RRE, rare reward event; RT, response time; sRPE, scalar reward prediction error; sRE, scalar reward expectation; TE, echo time; TI, inversion time; TR, repetition time; VOI, volume of interest; VS, visuospatial surprise; VTA/SN, ventral tegmental area and substantia nigra.

the outcome is better than expected, then there is a positive prediction error (PE), and the animal should revise its estimate of the choice's future value upwards. When the outcome is worse than expected—there is a negative PE—the animal should revise its future estimate of the choice's value downwards.

Neurophysiological investigations in animals have shown that areas such as the dopaminergic midbrain encode reward PEs [1, 3–7]. In addition, human functional magnetic resonance imaging (fMRI) studies have also examined whether activity throughout the brain reflects reward PEs [8–14]. While some neuroimaging studies have found evidence for PE coding in the dopaminergic midbrain, they have also reported PE coding in other structures, including striatum and prefrontal cortex. One aim of the current study was therefore to investigate the nature of PE coding in macaque prefrontal and cingulate cortex [15, 16] and compare it with that seen in other parts of the brain.

We also, however, intended to address a more general debate about the precise nature of the reward expectations that animals hold. While it is clear that they hold representations of average expected reward size, they also hold more specific representations about the nature and identities of the outcomes that they hope to receive after a choice [17–25]. If that is the case, then it might be anticipated that they would also experience PEs not just about the amount of reward—scalar reward value—but about other features of the reward [26].

It is possible that PE-related activity may actually reflect either scalar reward prediction errors (sRPEs) or PEs about other features of a surprising reward experience [27, 28]. In the current study, we disentangled 2 distinct types of reward PEs reflecting the richness of the reward representations held by monkeys. Specifically, we wanted to look at rare reward event (RRE) signals that might indicate that an unusual reward event has occurred. If such signals exist, then they would complement the sRPE signals typically studied. Alongside mean reward level, the frequency at which a given reward occurs is also important information that should be keenly monitored for learning purposes. In other words, animals may have a representation of the most frequent or modal reward levels, in addition to the mean reward level, to be expected from an average trial to guide their behavior [29]. Pierce and Hall [30] suggested that reward novelty is an important driver of the associability of any stimulus.

In the current experiment, it was possible to motivate behavior with primary reinforcers delivered directly to the animals' mouths. Because of the animals' training with relatively small but discriminable quantities, we could vary reward magnitudes parametrically and have a sufficient number of trials for animals to learn and change their reward expectations. To produce RREs, it was necessary to dissociate the reward frequency from absolute PEs. Therefore, we inverted the usual experience of receiving rewards close to the mean reward level most frequently. We did this by making actual instances of reward at the mean level very unlikely. This was achieved by devising schedules delivering mostly either 1 or 3 drops but occasionally delivering 2 drops. In such a schedule, the average reward expectation is close to 2 drops, but actual instances of 2 drops occurred only rarely. In other words, the reward distribution was a bimodal distribution with 2 equal peaks either side of the mean. Thus, while the delivery of a 2-drop reward would entail no sRPE, because it matched the average reward-amount expectation, it should constitute a large RRE because the delivery of 2 drops is such a rare and novel experience. Careful experimental design ensured that sRPEs and RREs shared only 0.049% of variance so that their neural correlates could be dissociated from one another.

However, as the world is full of minor or currently irrelevant changes, it is beneficial to selectively monitor reward frequency only when actively learning. Thus, we also examined whether increasing reward volatility (known to boost learning) [31, 32] also increases attention to other reward features such as RREs. We employed 2 types of schedules that we refer to as "changing/learnable" and "stable/unlearnable" (Fig 1A and 1B). Animals should learn to

expect either a higher or lower average reward in different parts of the changing/learnable schedules, potentially rendering an RRE more surprising or noteworthy because there is less uncertainty about whether to expect 1 or 3 drops of juice. By contrast, in the stable/unlearnable sessions, the 2 primary outcomes remain equally likely throughout, so no outcome is more frequent or expected and an RRE should be less surprising too.

In human neuroimaging studies, it is often difficult to determine whether all PE-related activity reflects encoding of the reward PE per se. This is because it is difficult to motivate human participants to perform tasks in the MRI scanner with primary rewards. Instead, task performance is typically motivated by visual cues that act as secondary, or other higher-order, cues that predict money given to the participant at the end of the experiment. It is therefore possible that apparent reward PE-related activity simply reflects the visual surprise associated with the appearance of a particular visual cue that is acting as a secondary reinforcer rather than the surprising level or nature of the reward it predicts [27, 33]. That visual surprise may be confounded with reward PE is an important consideration because it has been argued that dopaminergic neuron responses reflect the salience of cues and not just reward PEs [34]. While the salience of cues may be affected by many factors such as physical intensity, visuospatial surprise (VS)—appearance at a surprising location—may also contribute to salience. In addition to comparing sRPEs and RREs, we also considered whether reward PE and VS are encoded in the same manner by the same brain structures.

Specifically, we designed our experiment to concurrently measure the impact of VS and 2 distinct types of primary reward PEs. To infer levels of neural activity associated with these different error signals across the frontal cortex and striatum, we used fMRI. By studying macaques, it was possible to examine neural responses to PEs concerning primary reward that were of consequence and interest to the animals instead of visual tokens like those typically used in human neuroimaging. By carefully designing the order in which visual stimuli and rewards were delivered, not only did we decorrelate sRPEs and RREs, we also decorrelated both sRPEs and RREs from VSs. It was therefore possible to identify neural activity associated with each type of surprising event.

## Results

### Behavior

We wanted to investigate the behavioral and neural effect of spatial surprise and different types of reward surprise. We therefore designed an experiment that enabled us to look at 3 effects separately: sRPE, RRE, and VS (Fig 1C).

On each trial, animals were presented with a single blue rectangle appearing on either the left or the right side of the screen. VS could be examined because the common stimulus side reversed periodically, with occasional rare stimuli appearing on the opposite of the common location. Animals received a juice reward of 1, 2, or 3 drops after touching the response sensor on the side corresponding to the stimulus (Fig 1B). To first validate that monkeys can reliably distinguish between such juice amounts, we re-analyzed data from a different task and showed that it was indeed the case when both behavioral and neural data were analyzed (S1 Fig). The schedules were designed so that the average reward expectation across the session was for 2 juice drops to be delivered. Delivery of 3 drops or 1 drop therefore constituted positive and negative sRPEs, respectively (Fig 1C and 1D).

The juice rewards followed 4 different schedules that also allowed us to study the effects of RRE: during stable/unlearnable sessions (Fig 1A), the mean reward was constantly kept at 2 drops, but actually 2 drops of juice were only delivered in 10% of trials. This made receiving 2 drops of juice a surprising event, or RRE, even though it corresponded to the average expected

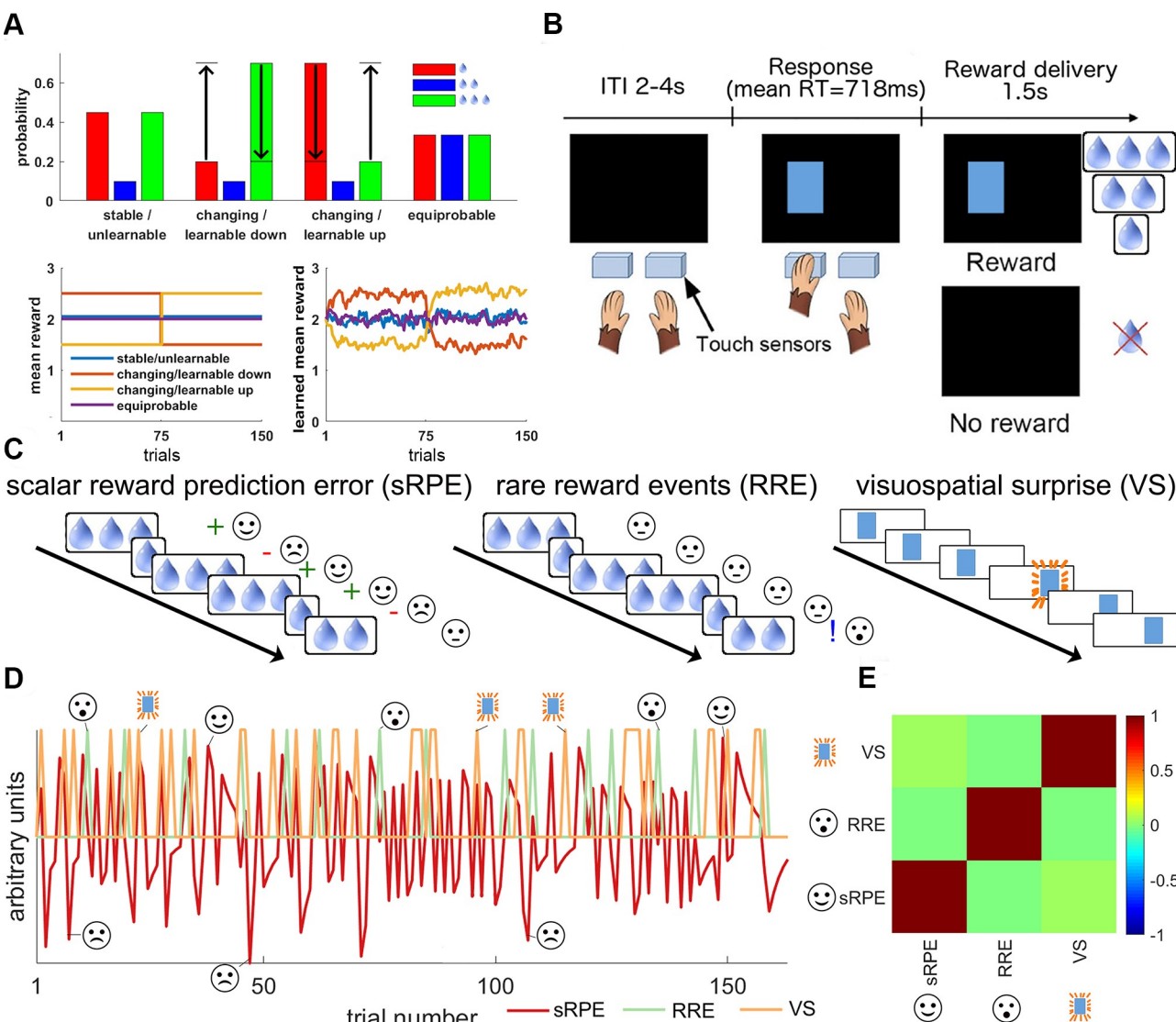

**Fig 1. Trial and task structure.** (A) Four reward schedules were employed. All had the same mean reward over the whole session, but in the learnable sessions (changing/learnable up; changing/learnable down) the average reward reversed halfway. RREs, which had the same scalar value, 2 drops, as the mean reward, only had a low probability of occurrence, P(RRE) = 0.1, except in the equiprobable condition where they were just as likely to occur as 1- and 3-drop outcomes (top panel). In changing/learnable sessions, monkeys have to learn that the average reward is higher/lower than expected at the beginning of a session and reverses halfway through the session (bottom panels). (B) On each trial, a visual cue (blue rectangle) appeared on either the left or the right of the screen. This instructed the monkey to make a hand response towards a touch sensor next to the corresponding side of the screen. If they responded correctly, then they received a juice reward of 1, 2, or 3 drops in size. (C) Illustrations of the 3 surprising effects of interest. sRPEs (left) occur when the obtained reward is better (i.e., 3 drops) or worse (i.e., 1 drop) than the expected average (2 drops). RREs (middle) occur when an infrequent reward is obtained (i.e., 2 drops). VS (right) occurs when the stimulus switches sides. (D) An example (stable/unchanging) session illustrating how sRPEs, RREs, and VSs occur on different trials. Trials on which each of the 3 types of surprise occur are marked. (E) Correlation matrix between the 3 main effects of interest showing that our task design allowed for separately examining the effects of sRPEs, RREs, and VSs. ITI, intertrial interval; RT, response time; RRE, rare reward event; sRPE, scalar reward prediction error; VS, visuospatial surprise.

reward for every trial. Five animals completed 6 of these sessions, and one animal completed 5 of these sessions. However, a weakness of this schedule is that reward monitoring and learning may be minimal or ineffective because the schedule is static and the reward environment is unchanging [31, 32]. We therefore created 2 further schedules that were more changing in nature—changing/learnable sessions—in which 2 drops were again delivered on only 10% of trials but the average reward changed up or down halfway through a session (Fig 1A). Once

again, the 2 drops of juice reward correspond to the average reward across the session (although the actual average was below or above depending on the part of the schedule; see Fig 1A lower panel). Once again because of their rarity, actual 2-drop reward occurrences are RREs. Each of our monkeys completed 4 of these sessions. We refer to these sessions as "changing/learnable" in contrast to the "stable/unlearnable" sessions because of 2 features they have: first, we expect monkeys to have formed strong priors about 2 drops of juice as the average reward amount because of the high number of stable/unlearnable sessions. Thus, when they encounter a changing/learnable session in which the mean reward amount starts at either 1.5 or 2.5 drops, they must re-learn average reward expectations. Moreover, once the mean reward changes halfway through a session, they yet again must re-learn average reward expectations (Fig 1A bottom right). The second reason we call these sessions "changing/learnable" is that the uncertainty about juice amount (1, 2, or 3 drops) that can be reduced in these sessions via learning is greater than in the stable/unlearnable sessions. This is because the inherent irreducible uncertainty about juice amount that is built into the schedules is greater for the stable/unlearnable than for the changing/learnable sessions: in the stable/unlearnable sessions, the monkeys can at best figure out that 1 and 3 drops of juice are equally likely (45% and 45%; Fig 1A top) and thus expect them with equal probability. In contrast, in the stable/unlearnable sessions, the monkey can figure out that on any trial the probability of one specific outcome is 70% and the other two outcomes are unlikely (10% and 20%; Fig 1A top) and thus form less uncertain expectations about what juice amount to expect.

Finally, we included a small number of sessions of an additional control condition during which receiving 1, 2, and 3 drops of juice was equally likely (Fig 1A); now 2-drop rewards are no longer RREs. Each of the 6 monkeys completed 2 of these sessions.

We wanted to assess whether factors related to VS, sRPE, and RRE affected behavior. To do so, we ran a generalized linear mixed-effect model (GLME1, Fig 2A; see Materials and Methods), predicting whether a performance lapse occurred—either an error response not directed to the correct side of the screen or an outlier response (any trial that had an unusually short [<50 ms] or long [>4,000 ms] response time [RT], indicative of task disengagement) on any given trial. Our 6 monkeys lapsed 15.95%, 15.17%, 11.80%, 10.24%, 9.89%, and 6.89% on average over all sessions they completed. It is intuitive that VS or negative/low sRPE events will be associated with performance lapses, but it is less clear that this will be true of RRE. It is, however, likely that performance lapses will diminish in frequency when scalar reward levels are high. We therefore combined the previously received rewards into a single regressor of scalar reward expectation (sRE) for the current trial through a Rescorla-Wagner learning model. Our results show that a positive sRE on the previous trial decreases the likelihood of a performance lapse on the current trial; in other words, greater reward expectation on the previous trial decreases the likelihood of a performance lapse on the current trial ($X^2(2) = 6.918$, $p = 0.031$; 2 degrees of freedom for fitting the learning rate and the inverse temperature; Fig 2A first column). To estimate the learning rate we used to calculate the sRE, we used another GLME (GLME2, Fig 2B) that included the reward on the previous 5 trials as individual regressors, and then estimated a Rescorla-Wagner type reinforcement learning model from these beta weights by finding the Rescorla-Wagner model that best describes the observed weights on previous rewards. We thus obtained a learning rate (alpha = 0.257). This procedure is broadly equivalent to fitting a truncated reinforcement learning model which also controls for other confounds (see Materials and Methods for details). To separately test for an effect of reward that decayed with the distance in the past (reward history), we also used GLME2 to fit a line through the beta weights of the previous 5 rewards for each monkey separately and then tested whether these slopes differ from 0. We indeed found a consistent effect of reward history (t(5) = 10.264, $p < 0.001$).

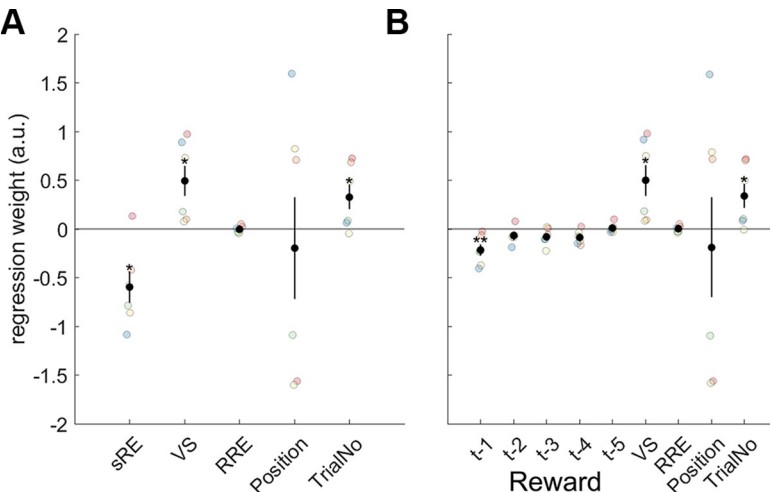

**Fig 2. Behavioral effect of surprising events.** GLMEs predicting errors or outliers as a function of 3 types of surprising event: sRE, VS, and RRE. The 2 GLMEs only differ in that, in A, sRE is indexed by a single regressor while in B, sRE is not a single regressor but instead its component parts are made explicit in terms of the reward outcome experienced on the last 5 trials (t-1 to t-5). Dots represent the beta weight associated with each regressor determined by a GLME applied to each monkey separately. Black dots indicate group mean effects according to the full GLMEs, and vertical bars indicate the SEMs. The Position effect indicates that animals were more likely to make errors or outlier responses depending on the side of the screen they had to respond to but that different animals had different side biases. Errors and outliers became more likely as each session progressed and TrialNo increased. Data and code to reproduce the figure can be found at https://doi.org/10.5281/zenodo.3993116. a.u., arbitrary units; GLME, generalized linear mixed-effect model; RRE, rare reward event; SEM, standard error of the mean; sRE, scalar reward expectation; TrialNo, trial number; VS, visuospatial surprise.

We did not find any significant effect of having experienced an RRE on the previous trial when all data were considered in aggregate ($X^2(1) = 0.004$, $p = 0.950$ in GLME1; third column in Fig 2A, seventh column in Fig 2B). However, we also performed these analyses for different session types (S2 Fig). While it is difficult to anticipate the impact, if any, that RREs might have on performance lapses, we noticed that RREs were more likely to be followed by performance lapses in the changing/learnable compared to the stable/unlearnable sessions in 5 of the 6 animals although the change occurred in the opposite direction in the sixth animal. A further set of GLMEs were used to predict RT as opposed to performance lapses (S3 Fig) and revealed VS slowed RTs ($X^2(1) = 9.409$, $p = 0.002$; GLME3). Finally, both GLMEs (GLME1 and GLME2) also revealed significant effects of VS: lapses of performance were more likely to occur if the stimulus appeared in an unexpected location ($X^2(1) = 5.837$, $p = 0.016$ in GLME1; second column in Fig 2A, sixth column in Fig 2B).

## fMRI

In our experiment, expectations could be violated in 3 main ways: VS, sRPE, and RRE. To identify brain areas associated with these 3 types of surprising events, we ran a 3-level multiple regression analysis by employing a general linear model (GLM). For each monkey, we used a fixed-effects model between sessions and applied the FMRIB's Local Analysis of Mixed Effects (FLAME) 1 + 2 procedure from the FMRIB Software Library (FSL) on the highest hierarchical level (level combining animals). We focus on effects on frontal cortex, striatum, and in the vicinity of the dopaminergic ventral tegmental area and substantia nigra (VTA/SN) in the midbrain because these areas are the ones that have been most frequently related to reward value expectation and PE coding in both human and nonhuman primates [2, 14, 35–40]. For this reason, we analyzed data in a volume of interest (VOI) covering frontal cortex and striatum

(S4 Fig) and a precisely localized region of interest (ROI) in the much smaller dopaminergic midbrain region VTA/SN. To further examine the co-occurrence of effects within the VOI, we place functional ROIs at peaks of the sRPE, RRE, and VS effects.

In this way, we attempted to ensure that we were able to detect surprise and PE responses wherever they occurred in the striatum or prefrontal cortex; such responses have previously been reported in several subregions in these structures; moreover, within the striatum it is not clear that the strongest PE/surprise signals can be mapped onto just the ventral striatum, caudate, or putamen. At the same time, the approach increased the statistical power of our analyses to examine neural activity in brain regions that were of a priori interest.

We anticipated that the VTA/SN's smaller size would preclude other analysis approaches commonly used in fMRI such as spatial cluster-based statistics that are most beneficial when there are large areas of activation; however, the activity in VTA/SN, as in the striatum and frontal cortex, was so prominent that it could also be identified using a standard cluster-based correction procedure for multiple comparisons corrected across the whole brain. Thus, in addition, whole-brain cluster-corrected results are reported in S5–S7 Tables and are discussed briefly in the Discussion.

We used data from 71 of the 76 sessions we had acquired (fMRI data from 5 sessions were corrupted and unrecoverable). For sRPE, VS, and RRE, our hypotheses necessarily focused on activity that was positively related to each type of surprising event, but we also note other patterns of activity where they existed.

To look for evidence of sRPE signals, we first performed an analysis in our VOI in the frontal cortex and striatum and in our ROI of interest in the VTA/SN. The analysis revealed 4 main clusters of activity (cluster $p < 0.05$, cluster forming threshold of $z > 2.3$; Fig 3A; see S1 Table for cluster locations). While this analysis approach—focused on a priori areas of interest —was our primary one, we note that the same results were evident in a whole-brain cluster-corrected analysis (S5 Table). Two of these clusters were located in the left and right ventrolateral striatum, respectively, and 2 in the left and right ventral sensorimotor cortex near the region occupied by the orofacial representation. Extracting the z-statistics of each session from the individual regressors of our whole-brain analysis from spherical ROIs with a 7.5-mm diameter in the left and right striatum, respectively, confirmed a clear effect of sRPE (Fig 3C). Next, we examined whether the same ROI carried information about VS or RRE. However, further analysis of the extracted z-statistics revealed no effect of VS in the left ($X^2(1) = 0.378$, $p = 0.539$; second column right panel Fig 3C) or right ($X^2(1) = 0.133$, $p = 0.715$; second column left panel Fig 3C) ventrolateral striatum. We also found no effects of RRE in the left ($X^2(1) = 0.485$, $p = 0.486$; third column right panel Fig 3C) or right ($X^2(1) = 0.023$, $p = 0.881$; third column left Fig 3C) ventrolateral striatum. Finally, we examined whether VS or RRE might exert a significant influence on activity but only in a specific session type (stronger learning effects are predicted in the changing/learnable sessions [31, 32]). This approach, however, failed to find any evidence for VS or RRE coding in striatum even in the changing/learnable sessions (S5 Fig).

To test whether an sRPE signal was present in the blood-oxygen-level-dependent (BOLD) activity in the ROI placed over the dopaminergic VTA/SN region in the midbrain, we warped an a priori–defined mask of the left and right VTA/SN into session space and extracted the z-statistics from the regressors of our whole-brain analysis for this ROI for each session (Fig 3B; although note, as already mentioned, that the effects were sufficiently strong to survive whole brain cluster correction; S5 Table; S6 Fig). We found an overall effect of sRPE in both the right and left VTA/SN, while controlling for the different hemispheres ($X^2(1) = 5.940$, $p = 0.015$; left columns in both panels of Fig 3D), but found no effects of VS ($X^2(1) = 2.187$, $p = 0.325$; central columns in both panels of Fig 3D) or RRE ($X^2(1) = 0.021$, $p = 0.885$; right columns in both

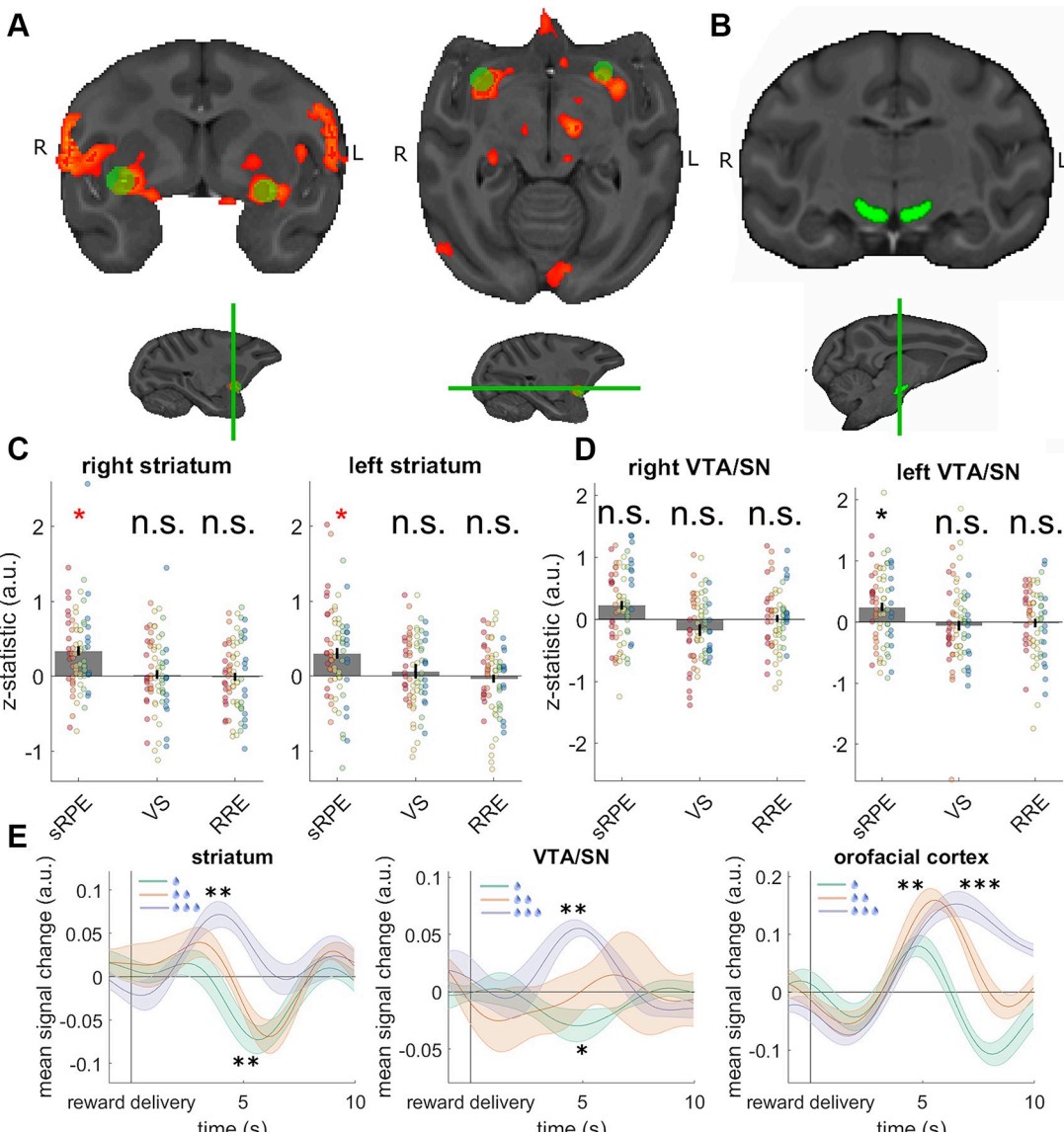

**Fig 3. sRPEs in the striatum and VTA/SN.** (A) Prominent sRPE effects were observed in the right and left orofacial somatosensory and motor cortex and ventrolateral striatum extending into the nucleus basalis of Meynert. (B) We used a priori ROIs in the right and left VTA/SN to extract z-statistics of the regressors from the whole-brain analysis for each session. (C) The z-statistics of each session from a spherical ROI placed at the peak activity of the cluster encompassing the right and left ventrolateral striatum. Different colors indicate data from different animals, with the grey bar showing the grand mean. A red asterisk indicates significance according to the whole-brain analysis, and a black asterisk indicates significance according to a test on the extracted z-statistics. The ROI-based analyses illustrated the presence of the sRPE effects in the left and right striatum but revealed no evidence for VS or RRE signals at the same locations. (D) The z-statistics for the ROI in the right and left VTA/SN revealed an overall significant effect of sRPE but no effects of VS or RRE. When testing the right and left VTA/SN separately, the z-statistics for sRPE in the left VTA/SN were significant, with no effects of VS or RRE. The right VTA/SN revealed a similarly signed sRPE effect although it was, on average, smaller in size and there was more variation across individuals and sessions. Once again VS and RRE effects were not significant in right VTA/SN. (E) BOLD time courses extracted from 3 ROIs in the ventrolateral striatum, the SN/VTA, and the orofacial cortex. The time courses are averaged over both hemispheres. For the location of the orofacial cortex ROI see S1 Fig. Time courses are illustrated for each level of juice reward the monkey received (1, 2, or 3 drops). The orofacial area activity reflects reward amount, and thus all three time courses exhibit an initial positive peak. In contrast, ventrolateral striatum and SN/VTA process sRPEs, and thus, the time course for receiving 1 drop of juice—which is associated with a negative sRPE—results in supressed activity (a negative activity change). Data and code to reproduce the figure can be found at https://doi.org/10.5281/zenodo.3993116. a.u., arbitrary units; BOLD, blood-oxygen-level-dependent imaging; n. s., not significant; ROI, region of interest; RRE, rare reward event; SN/VTA, ventral tegmental area and substantia nigra; sRPE, scalar reward prediction error; VS, visuospatial surprise.

panels of Fig 3D). When testing the right and left VTA/SN separately, we found an effect of sRPE in the left VTA/SN ($X^2(1) = 6.298$, $p = 0.012$; first column right panel of Fig 3D) although it did not reach significance in the right VTA/SN ($X^2(1) = 2.146$, $p = 0.143$; first column left panel in Fig 3D).

Finally, we considered the possibility that the sRPE effect might simply be artefact of the learning rate used when estimating monkeys' reward value expectations. In line with observations previously made by Wilson and Niv [41], a control analysis provided evidence for sRPEs in the VTA/SN regardless of the precise learning rate used (S7 Fig).

We found no effect of VS in the left ($X^2(1) = 0.635$, $p = 0.425$; second column right panel of Fig 3D) or right ($X^2(1) = 2.434$, $p = 0.119$; second column left panel Fig 3D) dopaminergic midbrain. We also found no effects of RRE in the left ($X^2(1) = 0.076$, $p = 0.783$; third column right panel of Fig 3D) or right ($X^2(1) = 0.025$, $p = 0.874$; third column right panel of Fig 3D) dopaminergic midbrain. Finally, we examined the effects of sRPE, VS, and RRE in specific session types (S5 Fig) in order to test whether VS or RRE effects might be present in the changing/learnable sessions in which learning effects were expected to be stronger; no evidence of their presence was found.

To illustrate these results, we extracted BOLD time courses from the ROIs in the ventrolateral striatum, the SN/VTA, and a control region in the orofacial cortex (Fig 3E; see S1 Fig for the location of the control region). When splitting up the time courses by the amount of juice received (1, 2, or 3 drops), we observe suppressed activity after monkeys receive 1 drop in the ventrolateral striatum ($X^2(1) = 10.219$, $p = 0.001$) and the SN/VTA ($X^2(1) = 6.250$, $p = 0.012$), which is due to the negative sRPE the monkeys experience. After receiving 3 drops of juice, which is associated with a positive sRPE, we observe enhanced activity in the ventrolateral striatum ($X^2(1) = 8.954$, $p = 0.003$, Fig 3E left) and the SN/VTA ($X^2(1) = 10.191$, $p = 0.001$, Fig 3E center). In contrast, an area that processes reward amount such as the orofacial sensorimotor cortex shows a different activity profile: we observe no effect different from baseline for 1 drop of juice because there is initially a positive BOLD response that is quickly followed by a negative change ($X^2(1) = 2.1.03$, $p = 0.147$), but we observe positive activity after monkeys receive 2 ($X^2(1) = 8.931$, $p = 0.003$) or 3 ($X^2(1) = 13.463$, $<0.001$) drops of juice (Fig 3E right).

Next, we examined the effects of RRE in an analysis conducted in our VOI in the frontal cortex and striatum and in our ROI in the VTA/SN. Note, as already mentioned, that careful experimental design ensured that sRPE and RREs shared only 0.049% of variance so that their neural correlates could be dissociated from one another. Moreover, by including terms relating to both sRPE and RRE in the same GLMs, we ensured that activity actually related to sRPE could not be misinterpreted as activity related to an RRE. No effects survived cluster corrections when we combined all session types at the contrast level. However, when we focused only on changing/learnable sessions, in which there was a possibility for animals to learn the changing statistics of the environment, we found a cluster of activation in the striatum extending to the posterior lateral orbitofrontal cortex (plOFC). The activity was situated lateral to the lateral orbital sulcus just anterior to the anterior insula in or near area 47/12o and extended dorsally towards the ventral tip of the arcuate sulcus in or near the proisocortical motor cortex (ProM) and area 44 (Fig 4A and 4B; S2 Table). The RRE-related activity in the striatum was situated in a more lateral and anterior area than was the case for the sRPE-related activity (Fig 3A). Again, this result was also apparent in a whole-brain cluster-corrected analysis (S6 Table). Extracting the z-statistics of the regressors of our whole-brain analysis from spherical ROIs with a 7.5-mm diameter placed at the peak activity of the cluster in striatum (Fig 4A) and a subpeak in plOFC (Fig 4B) illustrates the presence of the RRE effect in both areas (Fig 4C and 4D).

Further analysis revealed no such RRE signals in the striatum ($X^2(1) = 0.435$, $p = 0.510$, last column Fig 4C) or in the plOFC ($X^2(1) = 1.026$, $p = 0.311$, last column Fig 4D) during stable/

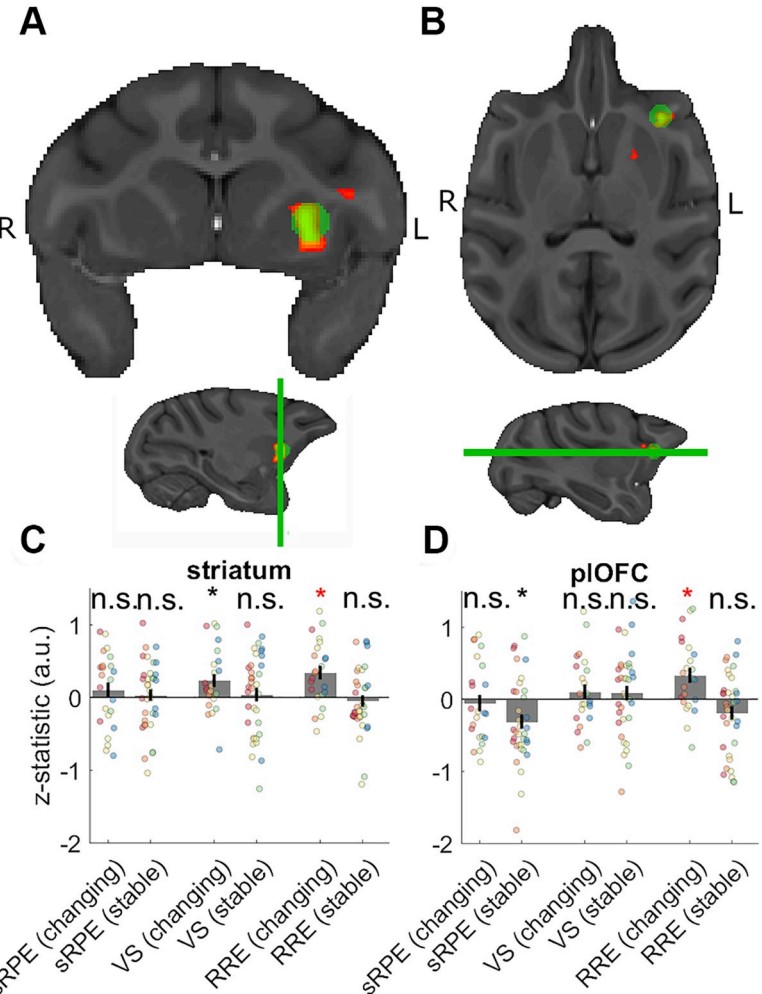

**Fig 4. Neural activity related to RREs.** (A) RRE effects in anterior lateral striatum during the changing/learnable sessions. (B) RRE effects in plOFC during changing/learnable sessions. The z-statistics of each session from a spherical ROI placed in (C) the striatum and (D) the plOFC. The ROI-based analysis illustrates the effects of RRE during changing/learnable sessions. Additionally, we found a significant positive effect of VS during changing/learnable sessions in the anterior lateral striatum. There was also a significant negative effect of sRPE during stable/unlearnable sessions in the plOFC although we are cautious about overinterpreting this result as it would not survive correction for multiple comparisons. A negative sRPE indicates a stronger response when reward outcomes are worse than expected. Data and code to reproduce the figure can be found at https://doi.org/10.5281/zenodo.3993116. a.u., arbitrary units; n. s., not significant; plOFC, posterior lateral orbitofrontal cortex; ROI, region of interest; RRE, rare reward event; sRPE, scalar reward prediction error; VS, visuospatial surprise.

unlearnable sessions. We then examined whether the same ROIs carried information about sRPE or VS (Fig 4C and 4D). We found evidence of a significant VS effect in the anterior lateral striatum ($X^2(1)$ = 7.729, $p$ = 0.005; Fig 4C) and a negative effect of sRPE in plOFC ($X^2(1)$ = 5.258, $p$ = 0.022; Fig 4D). We are cautious about overinterpreting the latter effect given that it would not survive correction for multiple comparisons.

Another way to test for effects of RRE is to examine whether activity during changeable/learnable sessions is significantly different from activity during equiprobable sessions. Running such an analysis on the whole-brain level revealed similar patterns of activity in the anterior lateral striatum and plOFC (S10 Fig; S11 Fig; S4 Table). In particular, our plOFC cluster during changing/learnable sessions remains broadly similar when contrasting RRE effects with

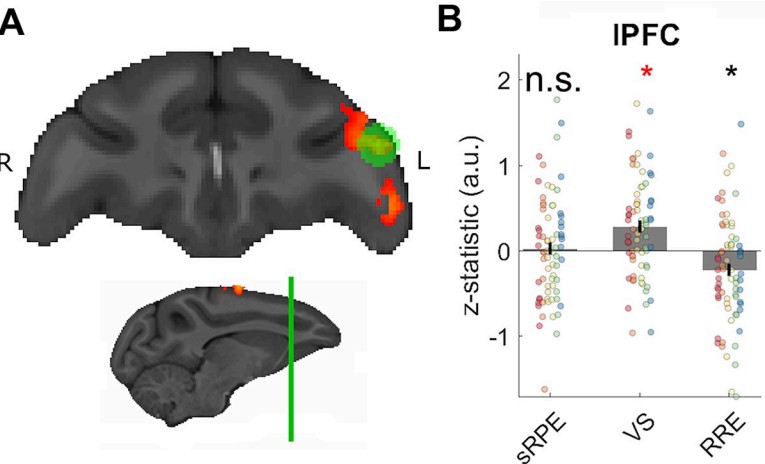

**Fig 5. Neural activity related to visual surprise.** (A) VS effects found in lPFC. (B) The z-statistics of each session from a spherical ROI placed at the peak activity of the cluster in lPFC. Labelling conventions are the same as in Fig 3. The ROI-based analyses confirmed the presence of the VS effects. There was no effect of sRPE. There was a significant negative effect of RRE, which is indicative of encoding of surprise about the scalar reward value (it corresponds to an unsigned reward PE). Data and code to reproduce the figure can be found at https://doi.org/10.5281/zenodo.3993116. a.u., arbitrary units; lPFC, lateral prefrontal cortex; n.s., not significant; PE, prediction error; ROI, region of interest; RRE, rare reward event; sRPE, scalar reward prediction error; VS, visuospatial surprise.

either of the other two conditions separately (changeable/learnable sessions versus equiprobable sessions and also changeable/learnable versus stable/unlearnable sessions) (S11 Fig).

Finally, we examined whether effects of VS were found at any other location in our VOI in the frontal cortex and striatum and in our ROI of interest in the VTA/SN. Four clusters were found. One cluster was found in lateral prefrontal cortex (lPFC; Fig 5A). Three other clusters were found just outside prefrontal cortex (the VOI was generously sized to include most of frontal cortex and adjacent tissue [S4 Fig] so that no prefrontal signals of potential interest were overlooked). These were situated in premotor cortex, secondary somatosensory cortex, and posterior cingulate area 23 (S3 Table). Again, the effect was strong enough to survive a whole-brain cluster-corrected analysis (S7 Table). Extracting the z-statistics of the regressors of our whole-brain analysis of each session from spherical ROIs with a 7.5-mm diameter illustrates the presence of the VS effect in lPFC (Fig 5B).

Next, we examined whether the same lPFC ROI carried information about sRPE or RRE. Extracting the z-statistics from the lPFC revealed no effect of sRPE ($X^2(1) = 0.120$, $p = 0.729$; first column Fig 5B) and no positive effect of RRE. There was, however, a significantly negative effect of RRE ($X^2(1) = 4.304$, $p = 0.038$; third column Fig 5B). A negative RRE effect is likely to indicate a response to a surprisingly large or small reward amount (sometimes referred to as an unsigned sRPE). Additional analyses of specific session types are reported in S9 Fig. It is perhaps worth noting here that while the aspect of VS we consider here might contribute to the salience of a stimulus, so might other factors. While no VS effect was found in the vicinity of the dopaminergic midbrain, it is quite possible that other aspects of visual stimuli contribute to the transient response that has been noted to occur at the onset of a visual stimulus associated with reward. Schultz, Lak, and Stauffer [36, 42] refer to this response to the onset of a reward-predicting stimulus as being due to the "physical impact" of the stimulus. By contrast, the VS effect that we examined compared the neural response to identical stimuli that were equally unexpected in time, but that differed because they were either in an unexpected or expected spatial location.

## Discussion

Animals, including humans, learn from the past to predict the future. New predictions are formed using PEs that occur during surprising experiences. There has been considerable interest in how PE's reward amount affects behavior and the manner in which they are encoded by activity in the dopaminergic midbrain [1, 2]. Here, however, we found behavioral and neural evidence for 3 distinct types of PEs in a group of 6 macaques.

We ran a relatively simple experiment in macaques in which they touched a rectangle appearing either on the left or right side of the screen, with correct responses delivering 1, 2, or 3 drops of juice. However, our key manipulations lay in the location of the target as well as the frequency of the juice amounts. Our schedules allowed us to examine the impact of visual PEs when the rectangle appeared on an unexpected side of the screen. Reward amount PEs (also known as sRPEs) were triggered when the amount of juice deviated from the average of recent experiences (as this regressor is signed, lower rewards led to suppression of activity, while more reward let to increased activity). The delivery of 2 reward drops conformed with the average reward amount, but at the same time it was the rarest outcome to occur, leading to a third form of surprising effect, i.e., RREs. The experimental design therefore allowed us to dissociate 3 types of surprising events, VSs, sRPEs, and RREs, and uniquely link them to changes in behavior and brain activity.

The 3 types of events were associated with different effects on behavior. Macaques were more likely to have a lapse in performance following VS (they either made an error response to the wrong side of the screen or an outlier response with a speed outside the usual range). By contrast, they were less likely to lapse after a high scalar reward value (positive effect of sRE). The novel reward experience on RRE trials (receiving 2 drops), however, while rare and thus surprising, did not significantly affect behavior in this very simple visual response task.

We did find consistently distinct neural responses for all 3 types of surprising events (S12 Fig). sRPEs significantly modulated activity in the vicinity of the dopaminergic midbrain consistent with previous neurophysiological studies [1, 3–7] and human neuroimaging studies [8–14]. There was no evidence that VS or RRE significantly altered activity in the same region. The findings can be linked to those reported by Lak and colleagues [43], who examined macaque dopaminergic midbrain neuron responses to rewards that varied in type, amount, and riskiness. They showed that activity reflected PEs for the integrated subjective reward value of the outcome. In other words, when multi-attribute reward outcomes were experienced, the contributions of the attributes on the scalar, subjective reward value also determined the PE response.

Several other studies have examined whether the activity of dopaminergic neurons or the BOLD signal in the dopaminergic midbrain responds to changes in the type of reward experienced even when its scalar value remains the same [44–47]. These studies have highlighted important similarities in the way in which the dopaminergic midbrain responds to both value and reward identity PEs. For example, across participants, there are correlations in the sizes of responses to value and identity PEs [44]. The RREs that we studied here are similar in some respects to identity PEs. They are both changes in the nature or type of reward rather than its value. However, the identity PEs investigated in previous studies reflect changes in features of the reward outcomes that are intrinsic to the outcomes themselves. By contrast, the RREs that we study here are neither changes in reward value nor changes in a taste or odor identity feature intrinsic to the reward itself. Instead, RREs are events that are only surprising in the context of knowledge of the composition of the current reward environment. It is only if the monkeys represent this aspect of the task space they are navigating that they will detect the occurrence of RREs.

The absence of a VS response in the current study is also interesting. In theory, it is possible that if the dopaminergic midbrain responds to surprising sensory events such as surprising visual stimuli then this could underlie the dopaminergic response to identity PEs. It has been reported that the human dopaminergic midbrain is responsive to surprising visual identities in a study employing face and house stimuli [48, 49]. On the other hand, it has been recently reported that reward identity PE responses in the dopaminergic midbrain are not proportional to the perceptual dissimilarity between the different rewards [45], suggesting that reward identity PEs are computed in some more abstract reward space.

There are 3 ways in which we might interpret the current failure to find VS responses in the dopaminergic midbrain. One possibility is that the absence of VS responses reflects the difficulty of imaging a small brain region such as the dopaminergic midbrain. From this perspective, it is important to remember that absence of evidence for an effect is not tantamount to evidence of absence of the effect. Against such considerations, however, it would also be important to note that VS effects were found in the current study in several cortical regions that correspond with those previously seen to be responsive, in humans, to surprising visual stimulus identities and that our imaging techniques were sensitive enough to identify sRPEs in the dopaminergic midbrain [48, 49]. A second possibility is that there is a fundamental difference between the VS we have studied here and the visual identity surprise studied in previous investigations [48, 49]; maybe the dopaminergic midbrain is responsive to the latter but not the former. Finally, it is possible that human participants in the previous study obtained some pleasure or satisfaction in correctly predicting the visual stimuli they were about to see even though, as in the current study, these were carefully decorrelated from reward events. It has certainly been argued that human participants are often motivated by a desire to perform a task well and are engaged by some of the task's design features as much or more than by the monetary rewards they will receive at the end of the experiment [50]. From this perspective, it would seem that when this level of task engagement is lacking, as in the monkeys in the current study, no dopaminergic response to VS is observable.

Dopaminergic neurons in the midbrain have been reported to respond to salient stimulus events in addition to responding to their reward value and reward value PE associated with a stimulus [42]. This has sometimes been described as a response to the "physical impact" of the stimulus or to the stimulus' capacity for "behavioral activation" as opposed to its precise scalar reward value [36]. In theory, the type of VS we investigated here might contribute to a stimulus' impact and capacity for behavioral activation. However, it also important to note that our experimental design and VS analysis controlled for other basic features of the visual stimuli we studied. VS-related activity was identified by comparing neural responses to visual stimuli of the same brightness, in the same locations, and with the same temporal predictability; only the predictability of the stimulus location differed between surprising and unsurprising VS events. Secondly, we also controlled for the "physical impact" of our visual stimuli in our GLMs by including a constant term at stimulus presentation but did not examine this effect further. As in the study conducted by Lak and colleagues, however, this analysis approach in the current study means that sRPE in the VTA/SN cannot be explained away as being the consequence of visual surprise.

Similarly, we found that sRPE, but not RRE or VS, was associated with activity in the ventrolateral striatum, suggesting that ventrolateral striatum also mostly coded for the current aggregate average value expectation. Thus, although animals may detect surprising reward events that are not sRPEs, they appear to employ neural mechanisms beyond the dopaminergic midbrain or ventral striatum to do so.

By contrast, RREs were associated with a very different pattern of activity in more lateral parts of striatum and plOFC. The activity was situated lateral to the lateral orbital sulcus in or

near area 47/12o and extended dorsally towards the ventral tip of the arcuate sulcus in or near areas ProM and 44. Interestingly, RREs were only tracked in sessions with changing average reward, i.e., when active monitoring and learning about the reward environment was more likely to occur [31, 32]. In this learning context, surprising events such as RREs might provide important information about a new state [51, 52]. On the other hand, in the stable/unlearnable sessions, animals cannot learn to expect either high or low reward levels because 1-drop and 3-drop outcomes are equally likely. Just as neither 1-drop nor 3-drop outcomes provide important information about reward state and so will be disregarded as uninformative, so too will RRE outcomes also be disregarded.

It has been argued that OFC is particularly important for reward learning and credit assignment [53]. OFC has also been proposed to represent task structure and to locate one's current latent task state within that structure. However, whether and how OFC decides when to encode a new task relevant state has been unclear [53]. Our data suggest that plOFC is specifically sensitive to novel reward experiences, or new state creation, when an agent has to consider what state they are currently in.

Although there has, as yet, been little investigation of the neural basis of RREs, the presence of an RRE signal in plOFC is consistent with the fact that it is known that the OFC does not just hold representations of reward size but rather representations that specify other features of the reward outcome [18–23]. The current results therefore suggest that, while it is important for animals to detect when rewards are unexpected in terms of a one-dimensional scale or common currency such as subjective value, equally animals track additional information about whether the nature and type of reward is surprising. Other authors looking for evidence of activity when surprising types of reward occur have also noted activity in the OFC [44, 45]. It is possible that evidence that the dopaminergic midbrain encodes a broader range of surprising events will emerge in future studies employing other types of surprise, interactions between different types of surprise, other behavioral paradigms, or other recording techniques. Nevertheless, the current results suggest the possibility that such forms of surprise coding may be present, or even more prominent, beyond the dopaminergic midbrain in regions such as the plOFC and perhaps elsewhere in the frontal cortex.

The plOFC not only tracked RREs but it did so in a relatively selective manner. The same plOFC region did not respond to VS. In addition, there was no positive response to sRPEs. However, it is not the case that VS had no impact on brain activity. Within the prefrontal cortex, VS was most prominently associated with activity in the posterior lPFC. This may be consistent with claims that the lPFC is part of a network responding to unexpected state transitions even when they do not have immediate implications for reward [54, 55]. In addition, outside the frontal lobe mask in which we conducted our analysis, exploratory analysis also revealed VS effects in the intraparietal sulcus and inferior parietal lobule. A related pattern of activity in the parietal cortex has been reported in fMRI studies of visuospatial attentional shifting in macaques [56, 57].

In addition, the lPFC exhibited a pattern of negative activity change in relation to RREs. This means that it is most active when either 1- or 3-drop rewards were received; in other words, lPFC activity is decreased when RREs occur. Instead, the lPFC responds to any reward magnitude deviating from the average expectation regardless of whether it is larger or smaller than the average [58]. Thus even though the lPFC may not detect RREs, it appears to encode both surprising reward amounts and surprising visual events, as might be expected from a domain-general learning mechanism [33].

We note that we were able to demonstrate specific PE and surprise effects because each region that was identified as showing a significant effect of sRPE, RRE, or VS was then examined to see whether it also held either of the two other types of activity patterns. Conducting a

test in this way provides a strong demonstration of specificity if the secondary tests fail to find other PE/surprise effects even when not correcting for multiple comparisons. We refrained, however, from comparing activity patterns in different areas statistically in case it might be argued that such comparisons are circular given that the areas usually had been first identified by their response in relation to a particular statistical contrast.

In summary, different types of surprising reward and visual events are associated with distinct effects on behavior and distinct neural circuits. Even within the domain of reward learning, there are distinct mechanisms linked with learning—on the one hand—about reward amount and its subjective value [43] and—on the other hand—mechanisms for learning about the precise features of potential rewards (such as their frequency) and for detecting RREs. While learning about reward value is associated with the dopaminergic midbrain and some divisions of the striatum, learning about other reward features and detecting RREs is associated with the OFC as well as with other divisions of the striatum.

## Materials and methods

Six rhesus monkeys (*Macaca mulatta*, 1 female) participated in the experiment. The animals weighed between 8.6 and 13.5 kg and were 7–8 years of age. They were kept on a 12-h light-dark cycle, with ad libitum access to water for 12–16 h after testing and throughout the day on nontesting days. All procedures were conducted under licences from the United Kingdom Home Office in accordance with the UK Animals (Scientific Procedures) Act 1986 and by the University of Oxford Animal Care and Ethical Review Committee.

In the behavioral task, monkeys sat in the sphinx position in a purpose-built MRI-safe chair (Rogue Research, Petaluma, CA). In order to prevent head movements during fMRI data acquisition, an MRI-compatible cranial implant (Rogue Research) was surgically implanted under anaesthesia.

Each trial began with a blank screen (2–4 s, mean 3 s) followed by a presentation of the rectangle. The monkeys responded by touching either of 2 custom-built infrared sensors placed in front of them. Each manual response was classified as either correct (the monkey touched the response sensor adjacent to the stimulus) or incorrect (the monkey touched the other sensor). Each correct response yielded a juice reward of 1, 2, or 3 drops (approximately 1.5 ml each drop) after a delay of 200 ms. The juice delivery took 1.5 s, and after an intertrial interval of 2–4 s (mean 3 s), the next trial began (Fig 1A). If the response was incorrect, the trial was repeated until the monkey made the correct response. Importantly, the spatial cue position (left or right) varied independently of reward magnitude (1, 2, or 3 drops) that was given for each correct trial. Reward was thus decorrelated from spatial position of the stimulus.

The task design enabled us to examine VS because the side on which the stimulus was shown reversed after 11–19 trials on the same side (mean 15 trials) although occasional stimuli appeared on the opposite side throughout. Eleven to 13 sessions of 150 rewarded (i.e., correct) trials were collected for each of 6 animals while they were in the MRI scanner. Thus, we could examine VS by comparing trials on which the stimulus had appeared on the same or the opposite side compared to the previous trial. Three types of sessions were performed by the monkeys, as follows.

### Stable/Unlearnable sessions

In 6 sessions, 1 and 3 drops were delivered randomly in 90% of the reward trials (45% for each reward size), and 2 drops were delivered in 10% of rewarded trials (Fig 1B, shown in blue). Mean reward over a session was kept at 2 drops. However, even if 2-drop rewards accorded with the average reward expectation, they were only rarely delivered, and so they were in this

sense the most surprising outcomes. Therefore, it was possible to identify activity related to RRE by comparing 2-drop reward outcomes with all other outcomes, and it was possible to identify sRPE-related activity by calculating the parametrically varying sRPE associated with each outcome. The sRPE and RRE regressors shared only 0.049% of variance. One weakness of the schedule, however, is that it is static and does not change over the course of the session. Because no learning is possible in such situations, learning mechanisms may not be deployed. We attempted to remedy this deficiency by using additional schedules.

## Changing/learnable sessions

In 2 different reward schedules (each comprising 2 sessions), the mean reward changed either from an average of 1.5 drops to an average of 2.5 drops ("changing up" sessions; Fig 1B, yellow line) or in the opposite direction, from 2.5 drops down to 1.5 drops ("changing down" sessions; Fig 1B, red line) halfway through the session. In these changing/learnable sessions, either 1 or 3 drops were delivered on 90% of trials on average, across the whole session. Therefore, once again it was possible to identify activity related to RRE by comparing 2-drop reward outcomes with all other outcomes, and it was possible to identify sRPE-related activity by calculating the parametrically varying sRPE associated with each outcome. However, it is possible that effects may be stronger in the changing/learnable sessions than the stable/unlearnable sessions because the reward environment is genuinely getting either better or worse. In order to estimate which is the case, animals must pay attention to outcomes. In these sessions, the sRPE and RRE regressors shared 0.1551% of the variance.

## Equiprobable sessions

In a third, control condition (comprising 2 sessions), we kept the average reward stable, but each reward magnitude had an equal probability of 1/3, thereby eradicating any reward frequency effects (Fig 1B, shown in purple). Therefore, once again it was possible to identify sRPE-related activity by calculating the parametrically varying sRPE associated with each outcome. Now, however, there may be less RRE effect because 2-drop reward outcomes are no less frequent, and therefore no more surprising, than 1- or 3-drop outcomes.

## Behavioral analysis

We fitted GLMEs to assess the impact of task manipulations on behavior. As explained, we were interested in the effects of VS, sRPEs, and RRE on behavior. Our binary regressor for VS encoded whether the stimulus appeared on the same side as on the previous trial or not. For this regressor, we zeroed out trials after the monkey had made an error, as these trials were repeated in our task design and would thus never lead to a VS. Our binary regressor coding for whether the current trial was a (surprising) 2-drop event (RRE) was also zeroed out for trials following an error. To calculate sRPEs, we first needed to compute the expected reward on each trial. To do so, we first had to establish the degree to which outcomes from previous trials contributed to the reward expectation that animals would hold on the current trial. To do this, we ran a GLME that included the rewards of the previous 5 trials as regressors. We then calculated the learning rate that best described the fitted beta weights of these 5 regressors and used it to calculate the expected reward (sRE) on each trial. Other regressors we included as confounds in our GLMEs were the position of the target on the screen (left or right) to account for a potential bias toward one side, and the trial number to account for time-on-task effects.

We used this GLME to predict lapses of performance (whether any trial was an error or an outlier with a very long RT). Both errors and outliers were coded in the same way in this analysis. We first marked all trials with an RT over 4,000 ms (indicating disengagement from the

task) or under 50 ms (indicating impulsive responding also consistent with a failure to engage with the task) as outliers. Additionally, trials with an RT of more than 2.5 standard deviations away from the mean were also marked as outliers.

In our GLMEs, we included random slopes for every monkey and a random intercept for every session. The GLMEs we ran, assuming a binomial distribution and using a logit link function, were (in Wilkinson-Rogers notation) as follows:

GLME1

ErrorOrOutlier ~ 1 + VS + sRE + RRE + Position + TrialNumber + (1 + VS + sRE + RRE+ Position + TrialNumber | Monkey) + (1 | Monkey:Session)

GLME2

ErrorOrOutlier ~ 1 + VS + Rewardt-1 + Rewardt-2 + Rewardt-3 + Rewardt-4 + Rewardt-5 + RRE+ Position + TrialNumber + (1 + VS + Rewardt-1 + Rewardt-2 + Rewardt-3 + Rewardt-4 + Rewardt-5 + RRE+ Position + TrialNumber | Monkey) + (1 | Monkey:Session)

The learning rate to compute sRE in GLME1 was obtained from GLME2. This was done by minimizing the Euclidean norm between the weights of the previous 5 rewards of a truncated Rescorla-Wagner reinforcement learning model (described by 2 free parameters: a learning rate and an inverse temperature) and the regression coefficients for Rewardt-1 to Rewardt-5 in GLME2. This procedure allowed us to both fit a learning rate while simultaneously accounting for other experimental factors in the linear model.

Additionally, we also ran each of these GLMEs separately for session types (i.e., stable/ unlearnable, changing/learnable, and equiprobable) and for each animal. Moreover, as a further control, we ran the same GLMEs with RTs instead of performance lapses as the dependent variable, assuming a gamma distribution and using the log link function. For these RT GLMEs, we either excluded all outlier trials (GLME3 and GLME4; see S3 Fig panels A–B) or all outlier, error, and repeat trials (trials after an error were repeated and thus did not result in a VS) (GLME5 and GLME6; see S3 Fig panels C–D).

To test for the significance of individual regressors in our GLMEs, we fitted the models once with and once without the regressor in question and performed a likelihood ratio test between the 2 models.

## MRI data acquisition and preprocessing

Imaging data were collected using a 3T clinical MRI scanner and a four-channel phased-array receive coil in conjunction with a radial transmission coil (Windmiller Kolster Scientific, Fresno, CA). For each monkey, structural images were acquired under general anesthesia, using a T1-weighted MP-RAGE sequence with a resolution of $0.5 \times 0.5 \times 0.5$ mm, repetition time (TR) = 2.05 s, echo time (TE) = 4.04 ms, inversion time (TI) = 1.1 s, and flip angle of 8˚. Two or 3 structural images per subject were averaged. Anaesthesia was induced by intramuscular injection of 10 mg/kg ketamine, 0.125–0.25 mg/kg xylazine, and 0.1 mg/kg midazolam. fMRI data were collected during task performance with a gradient-echo $T2^*$ echo planar imaging (EPI) sequence with a resolution of $1.5 \times 1.5 \times 1.5$ mm, interleaved slice acquisition, TR = 2.28 s, TE = 30 ms, and flip angle of 90˚. At the end of each session, to aid image reconstruction, a proton-density-weighted image was acquired using a gradient-refocused-echo (GRE) sequence with a resolution of $1.5 \times 1.5 \times 1.5$ mm, TR = 10 ms, TE = 2.52 ms, and flip angle 25˚.

EPI data were prepared for analysis following a dedicated nonhuman primate fMRI processing pipeline [59] using tools from FSL [60], Advanced Normalization Tools (ANTs) [61], and the Magnetic Resonance Comparative Anatomy Toolbox (MrCat; https://github.com/ neuroecology/MrCat). In short, after EPI data were reconstructed offline using a SENSE algorithm [62], time-varying spatial distortions were corrected using restricted nonlinear

registration, first to a session-specific high-fidelity EPI, then to each animal's T1w structural image, and finally to a group-specific template in CARET macaque F99 space [63]. Functional images were temporally filtered (high-pass cutoff at 100 s) and spatially smoothed (using a 3-mm full width at half maximum Gaussian kernel).

## fMRI analysis

Whole-brain analysis was conducted using a hierarchical GLM approach. Specifically, we first fitted every session individually before combining them for each monkey on a second hierarchical level using fixed effects in F99 standard space. Finally, we combined the data from all monkeys on a third hierarchical level using the FLAME 1 + 2 procedure from FSL [60] and using standard cluster-based thresholding criteria of $z > 2.3$ and $p < 0.05$ cluster-corrected [64]. Sixteen regressors of interest were designed for each session. Additional confound regressors were used to index head motion and volumes with excessive noise. All regressors were z-score normalized, and the data were analyzed using FSL's fMRI Expert Analysis Tool (FEAT). To model the hemodynamic response function (HRF), each regressor was convolved with a single gamma function (mean lag = 4.5 s, standard deviation = 2 s, therefore peaking 3.5 s after the event, which is consistent with a faster HRF in macaques than humans). The analysis and cluster correction were run only in a predefined VOI covering the frontal cortex and striatum, as shown in S3 Fig.

We were interested in the main effects of VS, sRPE, and RRE. We included 2 regressors, coding for VS occurring when stimuli were presented on the left and right side of the monitor to account for nonlinear differences between the two sides. These regressors were then combined on the contrast level to allow identification of neural activity related to VS independent of precise location of the surprising event. Finally, we note that, as in the behavioral analysis, we zeroed out the VS regressor whenever an error occurred on the previous trial. This is because, after an error, the trial is repeated until the monkey answers correctly, making it not spatially surprising any longer.

To examine the neural correlates of sRPE, we included 6 regressors at the outcome phase of each trial: 1 regressor encoding the magnitude of the current reward (0, 1, 2, or 3 drops) and 5 regressors encoding the reward magnitudes on the 5 previous trials. These 6 regressors were combined on the contrast level by subtracting the previous 5 rewards (which determine the animal's expectation about reward on the current trial) from the current reward. The regressors encoding the previous 5 outcomes were weighted according to the learning rate we fitted to our behavioral GLME (thus the reward expectation on the current trial is more influenced by recent previous rewards and less influenced by more distant past rewards). Additionally, because we observed some activity in the orofacial sensorimotor and gustatory cortex that could reflect the cortical activity correlates of swallowing and tasting the juice from the most recent outcome, we also contrasted the current reward outcome against a reward expectation estimate based on the weighted outcomes of 4 previous trials but leaving out the most recent trial. RRE events (receiving 2 drops of juice) were encoded with separate regressors both at outcome (whether the current reward is 2 drops) and at decision (whether the last reward was 2 drops). Finally, we also included 2 constants, one at decision and one at outcome, as well as a regressor that controlled for hand movements registered by the infrared touch sensor into the GLM. The decision constant accounts for the neural impact of visual stimulus presentation or motor activation [36, 42]. The outcome constant accounts for the neural response to receiving reward, regardless of magnitude [36, 42]. Note that by including terms relating to both sRPE and RRE in the same GLMs, we can ensure that activity actually related to sRPE cannot be misinterpreted as activity related to an RRE.

After having identified clusters of activity for our effects of interest, we were interested in whether the same or different regions process VS, sRPE, and RRE. To this end, we transformed the locations of the peaks of activity of our whole-brain analysis into session space, following the nonlinear deformation field. There, we extracted the average z-statistics for spheres with a diameter of 7.5 mm centered at the warped peaks for all regressors of interest. For the ROI covering SN/VTA, we used a mask from a recently published atlas [65], which we warped into session space. When effects were illustrated with extracted BOLD time courses, these time courses were extracted from the same ROIs and were up-sampled by a factor of 10 using spline interpolation. To test for statistical significance, we average the time courses of each session over time (from 0 s to 10 s in Fig 3 and from 0 s to 15 s in S1 Fig) and compare the effect over sessions and monkeys against baseline while controlling for subject-by-subject differences by modelling monkeys as random effects.

## Supporting information

**S1 Fig. Neural and behavioral evidence that macaques can distinguish between juice amounts.** To validate that monkeys can reliably distinguish between juice amounts of 1.5 ml, we re-analyzed data from a visual discrimination task. In this task, monkeys ($n = 4$) had to select one of 2 displayed stimuli that were probabilistically rewarded with a juice amount between 1 and 10 drops (0.5 ml per drop). The color of the stimuli varied in shade between blue and green. (A) The proportion of times the left stimulus was chosen as a function of the difference in reward magnitude (in ml) between the left and the right stimulus. For the slope of the curve in the central region (between magnitude differences of −2 ml and 2 ml), we can see that discrimination performance improves by 9.89% on average per drop (solid line; $t(7) = 23.08$, $p < 0.001$). (B) The thresholded and cluster-corrected map of activity covarying with reward from a whole-brain analysis of this task (shown with a threshold of $z = 4.5$). The cluster shows prominent bilateral activity in the orofacial sensorimotor cortex. We placed spherical ROIs at the most posterior local maxima in both the left and right orofacial activity (i.e., in the somatosensory cortex indicated by green ROIs; F99 coordinates 22.6, −4.02, 11.1 and −23.1, −3.51, 11.1) and extracted the BOLD time courses. (C) The extracted time courses from the ROIs indicated in (B) split by the amount of juice received (from 0.5 ml to 5 ml in 0.5-ml intervals) after reward delivery (top) and averaged over a window of 15 s (bottom). BOLD activity becomes stronger the more juice monkeys received in both the right $\mathbf{X^2(1) = 20.38, \mathit{p}<0.001}$) and left $\mathbf{X^2(1) = 17.687, \mathit{p}<0.001}$) orofacial somatosensory cortex. (D) The extracted time course from the same (now a priori) ROIs for the present study, again split up by the juice amount the monkeys received (1.5 ml, 3 ml, and 4.5 ml). As can be seen, BOLD signals are larger the more juice the monkey receives in both the right $\mathbf{X^2(1) = 8.892, \mathit{p} = 0.003}$) and left $\mathbf{X^2(1) = 10.984, \mathit{p}<0.001}$) orofacial somatosensory cortex. We thus conclude that our monkeys can reliably distinguish between 1, 2, and 3 drops of juice and that distinct activity patterns associated with each outcome are available as inputs to any neural learning mechanism in the present study. The averaged BOLD time course for both the left and right orofacial area is also shown in Fig 3E in the main text. Data and code to reproduce the figure can be found at https://doi.org/10.5281/zenodo.3993116.
(PNG)

**S2 Fig. Behavioral differences of RRE between session types.** To assess whether behavioral RRE effects were modulated by session type, we ran GLME1 separately for changing/learnable, stable/unlearnable, and equiprobable sessions. The effect was not significant in changing/learnable sessions ($\mathbf{X^2(1) = 0.318, \mathit{p} = 0.573}$; left column), stable/unlearnable sessions ($\mathbf{X^2(1) = 0.150, \mathit{p} = 0.699}$; middle column), or equiprobable sessions ($\mathbf{X^2(1) = 0.134, \mathit{p} = 0.714}$; right

column). Data and code to reproduce the figure can be found at https://doi.org/10.5281/zenodo.3993116.
(PNG)

**S3 Fig. Behavioral effects on RTs.** We also ran GLMEs with RTs as the dependent variable that contained the same independent variables as GLME1 and GLME2. For GLME3 (A) and GLME4 (B), outlier trials were excluded. GLME3 and GLME4 only differed in that in GLME3, the rewards from the previous 5 trials are combined into the sRE using a learning rate estimated from GLME2 as described in the main text. In GLME3, VS was significant ($X^2(1) =$ **9.409, $p$ = 0.002**), whereas sRE ($X^2(1) = 0.433$, $p = 0.511$) and RRE ($X^2(1) = 0.185$, $p = 0.667$) were not. In GLME5 (C) and GLME6 (D), we excluded all outlier, error, and repeat trials (trial after an error). Again, in GLME5, VS was significant ($X^2(1) = 9.631$, $p = 0.002$), whereas sRE ($X^2(1) = 0.444$, $p = 0.505$) and RRE ($X^2(1) = 0.030$, $p = 0.862$) were not. Data and code to reproduce the figure can be found at https://doi.org/10.5281/zenodo.3993116.
(PNG)

**S4 Fig. VOI covering the prefrontal cortex and striatum.** As we were primarily interested in prefrontal cortex and striatum and the nature of their PEs, we specified a VOI covering both. All whole-brain analyses in the main text were carried out using this VOI.
(PNG)

**S5 Fig. Neural effects of sRPE by session type.** We examined the extracted z-statistics shown in Fig 3, here split up by session type. Labelling conventions are the same as in Fig 3. Data and code to reproduce the figure can be found at https://doi.org/10.5281/zenodo.3993116.
(PNG)

**S6 Fig. Whole-brain results for sRPE in the dopaminergic midbrain.** When running a cluster correction on the whole brain (without the VOI) for our sRPE regressor we also found activity in the dopaminergic midbrain. The precise location of these clusters can be found in S6 Table. Data to reproduce the figure can be found at https://doi.org/10.5281/zenodo.3993116.
(PNG)

**S7 Fig. VTA/SN sRPE signals do not depend on the learning rate.** To consider the possibility that sRPE signals might be an artefact of the learning rate we used when estimating the monkeys' reward value expectations, we examined the BOLD response in the VTA/SN ROI further. By running a linear regression on the extracted BOLD time course time-locked to reward delivery (with terms coding for a constant, sRPE, VS, RRE, and trial number), we can determine the effect size of the sRPE regressor when constructing it using different learning rates. Effect size is here defined as the averaged beta weights for regressions run on the 10 s after reward delivery. The red line indicates the learning rate we used in the main text (0.257). As can be seen, the empirical learning rate is close to the peak effect strength in VTA/SN, but the effect strength is positive throughout regardless of learning rate. Data and code to reproduce the figure can be found at https://doi.org/10.5281/zenodo.3993116.
(PNG)

**S8 Fig. Neural effects of RRE by session type.** The z-statistics shown in Fig 4 split up by session type, including the equiprobable sessions in anterior lateral striatum and plOFC. Labelling conventions are the same as in Fig 3. Data and code to reproduce the figure can be found at https://doi.org/10.5281/zenodo.3993116.
(PNG)

**S9 Fig. Neural effects of VS by session type.** We also examined the extracted z-statistics shown in Fig 5 split up by session type. Labelling conventions are the same as in Fig 3. Data and code to reproduce the figure can be found at https://doi.org/10.5281/zenodo.3993116. (PNG)

**S10 Fig. Comparing neural RRE effects between session types.** As an alternative to the analysis shown in Fig 4, we also examined RRE by comparing changing/learnable and stable/unlearnable sessions against the equiprobable control sessions. (A) On the whole-brain level, we found significant activity in the striatum for a contrast comparing changing/learnable and equiprobable sessions. (B) We also found activity in the plOFC for the contrast comparing changing/learnable and equiprobable sessions although now the activity was even more centered on the boundary with the anterior insula. All local maxima of the cluster are shown in S4 Table. (C) Extracting the z-statistics from an ROI placed at the peak activity illustrates the difference for RRE between changing/learnable and stable/unlearnable sessions in the striatum (third column from the right and second column from the right). We did not find a significant difference between stable/unlearnable and equiprobable sessions for RRE ($X^2(1) = 0.516$, $p = 0.474$). Our analysis revealed a significant difference between changing/learnable and equiprobable sessions for sRPE in the striatum ($X^2(1) = 7.071$, $p = 0.008$) but with a different sign. For sRPE, the difference between stable/unlearnable and equiprobable was not significant ($X^2(1) = 3.160$, $p = 0.076$). For VS, neither the difference between changing/learnable and equiprobable ($X^2(1) = 0.110$, $p = 0.415$) nor between stable/unlearnable and equiprobable sessions ($X^2(1) = 0.516$, $p = 0.472$) was significant. (D) In the lOFC, we again did not find a significant difference between stable/unlearnable and equiprobable sessions for RRE ($X^2(1) = 0.665$, $p = 0.415$). For sRPE, there again was a significant difference between changing/learnable and equiprobable sessions in the opposite direction ($X^2(1) = 4.007$, $p = 0.045$). The difference between changing/learnable and equiprobable sessions was not significant for sRPE ($X^2(1) = 2.289$, $p = 0.130$). For VS, again neither the difference between changing/learnable and equiprobable ($X^2(1) = 2.552$, $p = 0.110$) nor between stable/unlearnable and equiprobable sessions ($X^2(1) = 0.002$, $p = 0.965$) was significant. Data and code to reproduce the figure can be found at https://doi.org/10.5281/zenodo.3993116. (PNG)

**S11 Fig. Locations of neural RRE activity.** RRE activity in plOFC in changing/learnable sessions (red) is located in the same area as activity when we contrast changing/learnable and equiprobable sessions (blue) or changing/learnable and stable sessions (green). All effects shown here are at a threshold of 2 and are shown without applying any cluster correction. Data and code to reproduce the figure can be found at https://doi.org/10.5281/zenodo.3993116. (PNG)

**S12 Fig. Summary of the neural results.** (A) VS occurred when the stimulus is expected to occur on one side of the screen (thought bubble) but surprisingly appears on the other side of the screen. We found activity in the posterior lPFC in response to VS. (B) sRPEs occurred when the macaque experienced a reward level that was higher or lower than its reward expectation (thought bubble). We found activity in the dopaminergic midbrain and ventrolateral striatum in response to sRPE. (C) RRE occurred when an infrequent reward (2 drops) was sampled. We found no neural activity for RRE results in in stable/unlearnable sessions. This result might be because in such sessions less monitoring of reward occurrences takes place because occurrences of 1 or 3 drops cannot be predicted (thought bubble). (D) In contrast, we found activity for RRE in lateral striatum and plOFC in changing/learnable sessions. This might be because in such sessions the frequency of 1 and 2 drops of juice is actively tracked to estimate the current average reward rate, rendering obtaining 2 drops of juice a potential event

of interest that is encoded as a new task relevant state (thought bubble).
(PNG)

**S1 Table. Clusters, z-statistics, and coordinates for sRPE in the VOI.**
(XLSX)

**S2 Table. Clusters, z-statistics, and coordinates for RRE in the VOI in changing/learnable sessions.**
(XLSX)

**S3 Table. Clusters, z-statistics, and coordinates for VS in the VOI.**
(XLSX)

**S4 Table. Clusters, z-statistics, and coordinates for RRE in the VOI for the contrast comparing changing/learnable and equiprobable sessions.**
(XLSX)

**S5 Table. Clusters, z-statistics, and coordinates for sRPE in the whole-brain analysis.**
(XLSX)

**S6 Table. Clusters, z-statistics, and coordinates for RRE in the whole-brain analysis in changing/learnable sessions.**
(XLSX)

**S7 Table. Clusters, z-statistics, and coordinates for VS in the whole-brain analysis.**
(XLSX)

## Author Contributions

**Conceptualization:** Urs Schüffelgen, Nils Kolling, Matthew F. S. Rushworth.

**Data curation:** Jan Grohn, Urs Schüffelgen, Franz-Xaver Neubert, Jerome Sallet.

**Formal analysis:** Jan Grohn, Urs Schüffelgen, Nils Kolling.

**Funding acquisition:** Matthew F. S. Rushworth.

**Investigation:** Urs Schüffelgen, Franz-Xaver Neubert, Alessandro Bongioanni, Jerome Sallet, Nils Kolling, Matthew F. S. Rushworth.

**Methodology:** Jan Grohn, Urs Schüffelgen, Lennart Verhagen, Nils Kolling, Matthew F. S. Rushworth.

**Resources:** Alessandro Bongioanni, Lennart Verhagen, Jerome Sallet.

**Software:** Jan Grohn, Lennart Verhagen.

**Supervision:** Nils Kolling, Matthew F. S. Rushworth.

**Visualization:** Jan Grohn.

**Writing – original draft:** Jan Grohn, Urs Schüffelgen, Jerome Sallet, Nils Kolling, Matthew F. S. Rushworth.

**Writing – review & editing:** Jan Grohn, Matthew F. S. Rushworth.

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
