## [Editor Report · Decision Letter 0]

4 Dec 2019

Dear Dr Grohn, 

Thank you for submitting your manuscript entitled "Multiple systems in macaques for tracking prediction errors and other types of surprise" for consideration as a Research Article by PLOS Biology.

Your manuscript has now been evaluated by the PLOS Biology editorial staff, as well as by an Academic Editor with relevant expertise, and I am writing to let you know that we would like to send your submission out for external peer review.

Please re-submit your manuscript within two working days, i.e. by Dec 06 2019 11:59PM.

Kind regards,

Gabriel Gasque, Ph.D.,

Senior Editor

PLOS Biology

---

## [Decision Letter · Decision Letter 1]

28 Jan 2020

Dear Dr Grohn,

Thank you very much for submitting your manuscript "Multiple systems in macaques for tracking prediction errors and other types of surprise" for consideration as a Research Article at PLOS Biology. Your manuscript has been evaluated by the PLOS Biology editors, by an Academic Editor with relevant expertise, and by three independent reviewers. Please accept my apologies for the delay in sending the decision below to you.

The reviews of your manuscript are appended below. You will see that the reviewers find the work potentially interesting. However, based on their specific comments and following discussion with the Academic Editor, I regret that we cannot accept the current version of the manuscript for publication. We remain interested in your study and we would be willing to consider resubmission of a comprehensively revised version that thoroughly addresses all the reviewers' comments. We cannot make any decision about publication until we have seen the revised manuscript and your response to the reviewers' comments. Your revised manuscript would be sent for further evaluation by the reviewers.

Having discussed the reviews with the Academic Editor, we think the reviewer's points are highly valid and you should focus on the methodological questions - particularly reviewer 1's point 1, and reviewer 2's point 2 (second and third paragraphs), as well as reviewer 3's point about rareness. 

We appreciate that these requests might represent a great deal of extra work, and we are willing to relax our standard revision time to allow you six months to revise your manuscript. We expect to receive your revised manuscript within 6 months.

**IMPORTANT - SUBMITTING YOUR REVISION**

*NOTE: In your point by point response to to the reviewers, please provide the full context of each review. Do not selectively quote paragraphs or sentences to reply to. The entire set of reviewer comments should be present in full and each specific point should be responded to individually, point by point.

*Resubmission Checklist*

*Published Peer Review*

*PLOS Data Policy*

*Blot and Gel Data Policy*

Sincerely,

Gabriel Gasque, Ph.D., 

Senior Editor

PLOS Biology

REVIEWS:

Reviewer #1: This manuscript describes a non-human primate fMRI study focusing on neural responses to different types of prediction errors. The elegant experimental design dissociates signed reward prediction errors (sRPE), visuospatial surprise (VS) and responses to rare events (RRE). In line with previous human fMRI and animal recording studies, sRPE correlate with activity in the ventral striatum and midbrain. However, these areas did not respond to VS and RRE. In contrast, VS trials evoked activity changes in lateral PFC, whereas RRE evoked activity in the lateral striatum and posterior lateral OFC, but only in sessions in which the probability of the two frequent events changed across trials. 

The results described here are both important and interesting. The study is well-designed and executed, and the manuscript is very well-written. However, there are two potential issues with the experimental design that should be addressed. In addition, I do not think the discussion is currently providing a balanced review of the literature and the findings. 

Major points

1. The authors present behavioral evidence that subjects can distinguish between 1 and 3 drops of juice. However, there is no evidence that they can detect the difference between 1 and 2, and 2 and 3 drops, especially in the stable sessions. This has implications for the interpretations of the imaging findings. First, if subjects cannot detect this difference, there may be no brain response. This could explain the null results in the RRE condition, and thus, a failure of the experimental manipulation to induce the intended effect needs to be taken into consideration in order to avoid prematurely rejecting the H1 (i.e., the results could have come out very differently if the experiment had used 1, 3, and 5 drops). In addition, 2 drops could have been perceived as more ambiguous and thus responses to 2 drops may differ in other important aspects (e.g., lower confidence) which needs to be considered when interpreting these results. 

2. The design confounds sRPE with reward size. This is evident in the somatosensory responses observed in Fig. 3A. As the authors are well aware, one way to dissociate RPE from reward size is to compare responses to the same reward size between trials with different expectations. Because expectations in the current experiment were relatively constant across all trials (given the small learning rate), this may be difficult to achieve here. Nevertheless, the authors could estimate the effects of a parametric sRPE regressor that is orthogonalized with respect to reward size and test whether the results are similar. In addition, expectations change more strongly in the learnable sessions, so those sessions may provide another avenue to address this issue.

3. The authors conclude that "while the results here and in previous studies (Lak et al. 2014, PNAS) demonstrate that activity in the dopaminergic midbrain reflects PEs along a one dimensional scale of subjective value, unexpected reward feature information is not tracked." I do not think these conclusions are warranted, and I also don't think this statement reflects a balanced review of the literature. First, Lak et al. never actually tested responses of dopamine neurons when value-neutral expectations about juice type were violated. Second, value-neutral "sensory" prediction errors have recently been identified in the midbrain by other groups. For instance, in rats, the same dopamine neurons that respond to sRPE also respond to errors in reward identity predictions (Takahashi et al. 2017, Neuron). Similar responses to value-neutral identity prediction errors (Boorman et al. 2016 Neuron; Howard & Kahnt 2018, Nature Communications; Suarez et al. 2019, eLife) and low-level sensory prediction errors (Iglesias et al. 2013, Neuron) have previously been described in the human midbrain. Third, there is a real possibility for false negatives here (see point 1), and thus, interpretation of the current null results regarding RRE in the midbrain need to consider the possibility that 2 drops may not have been discriminated reliably from 1 and 3 drops. 

Reviewer #2: The goal of the manuscript is to determine whether canonical reward brain areas demonstrate similar neural signals for reward prediction errors (PEs) to other PEs, such as rare reward surprise and visual cue novelty that instead provide information regarding attributes of a stimulus rather than reward value. Monkeys performed a simple, but clever task designed to dissociate these three types of PE while undergoing fMRI scanning. The manuscript provides evidence for different areas of the brain responding to variables related to classic, scalar PE (striatum and VTA/SN), visual surprise (PFC), and rare reward (striatum & OFC). The manuscript addresses an important gap in the field regarding reward and stimulus prediction. However, there are details in the paper that need clarification before conclusions can be drawn from this work.

Major concerns

1. The current organization the manuscript make pieces of it difficult to follow. For example, the abstract very clearly delineates the goals and the variables that will be presented in the paper, but then that same structure is not recapitulated in the introduction to the same extent. It is not until the fifth paragraph of the intro that the authors introduce why they designed the experiment the way they did. It may be helpful to frame that more concretely sooner in the introduction and then follow that structure throughout the rest of the introduction, and the results. Similarly, at the end of the introduction (and abstract) the idea of "learning" is used without introducing or defining it. There are further comments about learning below.

Another organizational item is the order in which the figures are discussed. At several places in the manuscript the panels of the figures are introduced not in the order that they are presented. For example Figure 1B is first referenced in the introduction without having referenced Fig. 1A. In the results, Figure 3B is discussed before Figure 3A. In general, I think it would be helpful to introduce results in the brain as a whole before discussing results in ROIs.

2. It would be helpful to have more of the experimental details more up front. For example, the total number of sessions, runs, trials etc. for each trial type in each animal were not apparently reported. What was the error rate of each animal? Given that relative instances of each kind of trial are very important across the manipulations, establishing the actual numbers and frequencies based on the animal's actual performance of each I feel is important. References are made in the results to the average number of trials and sessions, but more specifics would be helpful.

A second important detail that needs more clarification is the use of the striatum and PFC large VOI. In my first read through of the results, it is was not at first clear how the VOI was being used. I think the manuscript should front this information much more strongly and be careful about the use of the phrase "whole brain". Also, for one analysis, of VS, the manuscript mentions one whole brain alternative analysis that is presented in the supplement (top of pg 16, Sup. Fig 6). In the discussion exploratory analyses showing IPS and IPL are mentioned, but I do not think those analyses were introduced in the results, nor are they shown in the supplement. I very strongly feel that even if the authors want to focus on the frontal cortex and striatum for their main analyses, that that whole brain analyses without the use of the large VOI should be shown, even if it's at lower threshold, in the supplement. It is difficult to contextualize how unique these results may or may not be to these regions without this information.

A third detail under this heading is the use of other ROIs in the paper. Sometimes they are independently-anatomically defined and sometimes they are defined based on peaks of clusters from the same contrasts they are used to examine. Given the logic presented in the manuscript for using the VOI (the strong a-priori hypotheses about frontal cortex and striatum), it was not clear to me why the authors did not use unbiased-anatomical ROIs throughout. Further, the authors currently group the striatum as a single structure - if anatomical ROIs are used then including CN, Put, NAc etc. and correcting for multiple comparisons may be a more straightforward approach.

3. There are many places throughout the manuscript where claims are made that one ROI showed responses to one variable while others didn't. What was not clear to me was whether there were statistical comparisons performed to determine if the representations were statistically greater in one area over another. Are there ANOVAs or similar tests that were performed to show that effects were different? Are there variables that can be directly contrasted against each other? The manuscript mentions one such contrast in Sup. Fig. 6 and perhaps one other, but nothing further. If the manuscript would like to conclude that there are differences in representation of these variables, then it would seem to be appropriate to directly test/contrast them against each other. Also, more comparisons to the equiprobable condition would bolster their claims.

4. I struggled with the design and interpretation of the changing/learnable condition. First, it was not clear exactly what was changing/learnable at first. The manuscript uses "reward rate" I believe to describe that condition, but that would seem to imply that the timing between the rewards would change to be more or less frequent. My interpretation from the methods is that it is not the timing between the rewards that changes, but the relative ratio of one drops or three drops that the monkey receives. Again, this is where trial counts and run structure would be helpful for understanding exactly what the monkeys are doing. A more detailed description of what the manipulation is in the introduction and/or results would be helpful. Was this manipulation in the same session, different sessions, how many?

The manuscript takes careful time to consider what is traditionally thought of as PE and how definitions could be different and how they test those differences. It would be helpful if the manuscript could give the same careful introduction and consideration to possible definitions of learning and what it means in the context of this task that the animals are doing. In some sense, all the conditions are "learnable" because they rely on the animal having a (learned) expectation of the stimuli and the drops of juice. Of course there are many kinds of learning, but often learning is thought of in terms of association, but what is changing in the learnable condition is not an association, but an expectation of the average amount of reward. It was not clear to me on what time scale this change was happening, and if the animal would have any reason to track this information or if there was any behavioral evidence that the animal used this change in information. There is some mention (pg 10) of responses to RREs being different in stable vs. changing, but I did not see further discussion of it.

Given all of the above, I felt more details were needed before being able to evaluate the striatum/OFC results for the changing condition. Might it also be true that neural and/or behavioral responses to sRPEs might be different when there are changing expectations?

Other concerns

1. There are many details that would be helpful for the clarity of the figures with brains & ROIs on them. 

a. An indicator of which side of the brain is R/L and the slice numbers for the slices shown I believe is not present in any of the figures. 

b. The large arrows pointing from the brains to the graphs below make it difficult to understand what volumes/ROIs are being used. For example, in Figure 3A the arrow goes from an area of univariate results where an ROI sphere was used (and it's not clear where the sphere sits in relation to the results shown), whereas in Fig 3B there is no arrow from the anatomical ROI. I would recommend removing the arrows and outlining the area used as an ROI.

c. The colored stars to indicate different kinds of significance are confusing.

d. The black dashed lines indicating the mean are very very difficult to see

e. I appreciate the authors desire to show all the data for each animal, but the many shapes and colors make it difficult to extract the important information from each graph. Perhaps the authors could experiment with other kinds of plots that still show the information (violin plots?) or different colors that show the detail if needed, but it is (visually) easier to background information that is not immediately the most important for each figure.

f. If the figures are going to distinguish between stable and changing conditions, then perhaps some analyses as to whether they are the same or different could be provided. It's hard to know if we should take anything from the triangles and squares because they are very hard to see.

2. There are mislabeled figure references in text for Fig 2. A and B- page 10. With reference to the columns- the order of the graphs is reversed: text states 'sixth column in Fig 2A,

second column in Fig 2B'. (Fig 2A only has 5 columns total)

3. There is reference to GLME5 and GLME6 based on RT analysis in the methods, but results and further reference to these are not mentioned with that section (pg. 10) or elsewhere.

Reviewer #3: This manuscript describes an fMRI experiment in rhesus macaques that examines the neural correlates of scalar reward prediction errors, visual surprise, and rare reward events. The latter to potentially account for how surprise might drive learning based on perceptual versus reward inference processes, and constitute the main conceptual advance of the manuscript. BOLD activity in the putative dopaminergic midbrain and striatum was correlated with RPEs (only in the left VTA) and not VS or rare reward events. Activation in lOFC was correlated with visual surprise (i.e. whether a cue associated with different amounts of reward appeared in a frequent versus rare location). Posterior lOFC was identified as encoding rare reward events, but in only in sessions where the mean reward rate drifted up or down over the course of the session. While interesting, several aspects of the experimental design and analyses raise concerns as to the reproducibility and robustness of the reported results. 

Fitting a value updating algorithm to the beta coefficients resulting from a general linear model fit to predict performance lapses (i.e. GLME2) to derive learning rates and sRPEs is a circular analysis strategy, because GLME2 already accounts for value integration by including the reward outcomes on the previous five trials. Instead the value updating algorithm should be used to estimate value updates directly with free the learning rate set as a free parameter to either predict performance lapses or reaction times. The authors could even use arbitrary learning rates, as it was recently demonstrated that model-fitting isn't necessary for identification of model-based fMRI results (Wilson & Niv, 2015). Do the authors obtain similar results when they estimate sRPEs directly based on the empirical reward value and an arbitrary learning rate?

For the ROI analyses were scalar RPEs, visual surprise, and rare reward event regressors tested separately or in the context of the same model? If they were not assessed simultaneously then a correction for multiple comparisons needs to be applied.

Dopamine neurons have been shown to code both visual novelty responses as well as reward prediction errors (Lak et al., 2016). Their initial phasic activity reflects novelty while a later component of their activity encodes RPEs related to learning. It is therefore surprising (or maybe not given the low SNR of BOLD activity in this region) that the authors would not find neural correlates of visual surprise in their VTA/SN analyses.

Of major concern is that the rare, rewarding event correlate in posterior lOFC only appears in the sessions in which the average reward rate drifts upwards or downwards. As a result of this manipulation the rare, rewarding events delivering 2 rewards now are both surprising but also elicit a prediction error. To accurately identify this region as encoding the rarity of this event versus RPE coding it is necessary to include these RPEs, no matter how small, in the GLM in addition to the rare reward regressor. Otherwise this is a fundamental flaw in the experimental design. Moreover, the lOFC region identified as a neural correlate of visual surprise and rare reward events appears to be driven by two individual monkeys in each case. The error bars for most of the animals contain 0 indicating the effect is not consistent across the animals. Compare this to the scalar RPE correlates observed bilaterally in the striatum, where the error bars for the majority of the animals do not contain 0. 

"Another way to test for effects of RRE is to examine if activity during changeable/learnable sessions is significantly different from activity during equiprobable sessions." This would also include a key comparison of whether the RRE effect in sessions in which the mean reward rate changes over time differs from sessions in which it remains stable, where an RRE was not detected in posterior lOFC. For Fig. 5 and Fig. S5 the authors should report on direct comparisons of these two session types, as well as comparisons to the equiprobable sessions. Also, it is curious as to why the voxel-wise comparisons of the changing and equiprobable sessions session type fails to identify the same posterior lOFC cluster as shown in Fig. 5.

The described effects are difficult to evaluate by the symbol and color coding scheme. The plotting conventions adopted in the Supplementary Figures, where the Z-statistics are broken out by session type, should be used in the main manuscript figures. It would also be helpful given that this is a within-subject design to link the plotted statistics across the sessions for each animal. 

The use of the mask for cluster correction as depicted in Fig. S3 is not well justified and an unconventional practice. Usually such masks correspond to smaller regions of interest (i.e. the VTA/SN) not entire swaths of the brain. Do the identified cortical and striatum clusters not survive more traditional whole-brain cluster correction? Is there some reason that this mask needs to be applied. For example, is the SNR highest in these brain regions?

While Figure 6 is a nice illustration that might be suitable as a graphical abstract it does not belong as a main figure.

---

## [Decision Letter · Decision Letter 2]

13 Aug 2020

Dear Dr Grohn,

Thank you for submitting your revised Research Article entitled "Multiple systems in macaques for tracking prediction errors and other types of surprise" for publication in PLOS Biology. I have now obtained advice from the original reviewers and have discussed their comments with the Academic Editor. 

We're delighted to let you know that we're now editorially satisfied with your manuscript. However before we can formally accept your paper and consider it "in press", we also need to ensure that your article conforms to our guidelines. A member of our team will be in touch shortly with a set of requests. As we can't proceed until these requirements are met, your swift response will help prevent delays to publication. Please also make sure to address the data and other policy-related requests noted at the end of this email.

*Copyediting*

*Published Peer Review History*

*Early Version*

*Submitting Your Revision*

Sincerely,

Gabriel Gasque, Ph.D.,

Senior Editor,

ggasque@plos.org,

PLOS Biology

ETHICS STATEMENT:

-- Please include the full name of the IACUC/ethics committee that reviewed and approved the animal care and use protocol/permit/project license. Please also include an approval number.

DATA POLICY:

Note that we do not require all raw data. Rather, we ask for all individual quantitative observations that underlie the data summarized in the figures and results of your paper. For an example see here: http://www.plosbiology.org/article/info%3Adoi%2F10.1371%2Fjournal.pbio.1001908#s5

These data can be made available in one of the following forms:

Regardless of the method selected, please ensure that you provide the individual numerical values that underlie the summary data displayed in the following figure panels: Figures 2AB, 3C-E, 4C-D, 5B and Supplementary Figures 1ACD, 2, 3A-D, 5A-D, 7, 8AB, 9, and 10CD.

Please also ensure that each figure legend in your manuscript include information on where the underlying data can be found, and ensure your supplemental data file/s has a legend.

Reviewer remarks:

Reviewer #1: The authors have adequately addressed my initial comments. The unpublished fMRI and behavioral data from a different experiment, and the analysis of fMRI signals in the orofacial cortex in the current study convincingly show that it is very likely that monkeys were able to distinguish between the 3 drops. In addition, adding responses to the 3 drops in the striatum, midbrain, and orofacial cortex is a nice way to show that the midbrain and striatum respond to prediction errors rather than reward magnitude. Finally, the revised discussion provides a more balanced interpretation of the findings.

Overall, I think the revised manuscript is much improved and I congratulate the authors to an important contribution. 

Reviewer #2: The authors have adequately addressed the reviewer comments by restructuring the organization of the introduction and figures to match for consistency, clarification of concepts and design implementation by providing information directly up front in the manuscript. The additional statistical analyses included in both the paper and the supplemental material clarify and bolster claims made in the paper and in response to concerns raised by the reviewers. Additionally, the figure clarity and details regarding statistical tests within the manuscript has been significantly improved. Together, these revisions greatly improve the content and strengthen the manuscript. 

This work contributes novel findings to our understanding of the processing of scalar PE, RRE, and VS within the macaque brain. 

Reviewer #3: The authors have thoughtfully addressed my previous concerns.

---

## [Editor Report · Decision Letter 3]

18 Sep 2020

Dear Dr Grohn,

On behalf of my colleagues and the Academic Editor, Ben Yost Hayden, I am pleased to inform you that we will be delighted to publish your Research Article in PLOS Biology. 

Early Version

PRESS 

Kind regards,

Vita Usova

Publication Assistant, 

PLOS Biology

on behalf of

Gabriel Gasque,

Senior Editor

PLOS Biology